# Suboptimal infant and young child feeding practices in rural Boucle du Mouhoun, Burkina Faso: Findings from a cross-sectional population-based survey

**Sophie Sarrassat**[1]*, **Rasmane Ganaba**[2], **Henri Some**[2], **Jenny A. Cresswell**[1], **Abdoulaye H. Diallo**[3], **Simon Cousens**[1], **Veronique Filippi**[1]

1 Centre for Maternal Adolescent Reproductive and Child Health, Department of Infectious Disease Epidemiology, London School of Hygiene and Tropical Medicine, London, UK, 2 Africsanté, Bobo-Dioulasso, Burkina Faso, 3 Centre Muraz, Bobo-Dioulasso, Burkina Faso

* sophie.sarrassat@lshtm.ac.uk

## Abstract

### Introduction

In Burkina Faso in 2016, 27% and 8% of children under-5 were estimated to suffer from stunting and wasting respectively. Here, we report on infant and young child feeding (IYCF) practices in rural areas of the Boucle du Mouhoun region.

### Materials and methods

A cross-sectional population-based survey was performed in 2017 in a representative sample of mothers of children aged 6 to 23 months. IYCF practices were assessed using 24-hour dietary recall. Logistic regression was used to identify predictors of IYCF practices. All analyses accounted for sampling stratification by child's age group and for data clustering.

### Results

According to mothers' reports, 60% (95%CI 55, 65%) of children received the minimum meal frequency, but only 18% (95%CI 15, 22%) and 13% (95%CI 10, 16%) benefited from the minimum dietary diversity and the minimum acceptable diet respectively. Only 16% (95%CI 13, 20%) of mothers reported increasing breastfeeding or liquids and continued feeding during an episode of child illness. Knowledge of timely introduction of complementary foods and recommended feeding practices during an illness were low. Despite positive attitudes towards the introduction of key food groups, mother's perceived self-efficacy to provide children with these food groups every day was relatively low.

### Discussion

Our findings highlight the need for interventions to improve mothers' knowledge and practices in relation to IYCF in the Boucle du Mouhoun region. Behaviour change communication strategies have the potential to improve IYCF indicators but should be tailored to the

**Data Availability Statement:** Due to ethical restrictions, the full dataset will be made available to bona fide researchers on condition that they

provide evidence of ethics approval and sign a Data Transfer Agreement. Further information on the data and access conditions can be found through the LSHTM Data Compass at: https://doi.org/10.17037/DATA.280.

**Funding:** Alive & Thrive is funded by the Bill & Melinda Gates Foundation and the governments of Canada and Ireland and managed by FHI 360. This research was supported by funding from the Bill & Melinda Gates Foundation, Grant Number OPP50838.

**Competing interests:** The authors have declared that no competing interests exist.

local context. The high attendance of health facilities for preventive well-baby consultations represents an opportunity for contact with caretakers that should be exploited for promotion and child growth monitoring.

## Introduction

In 2011, under-nutrition, consisting of foetal growth restriction, stunting, wasting, and deficiencies of vitamin A and zinc, along with suboptimal breastfeeding, was estimated to underlie about 3.1 million under-five deaths, corresponding to 45% of all deaths in this age group [1]. Improvements in nutrition will therefore be essential to the achievement of the Sustainable Development Goal (SDG) target 3.2 of 25 or fewer under-five deaths per 1,000 livebirths by 2030. To this end, the SDG targets 2.1 and 2.2 call respectively "*to end hunger and ensure access by all people, in particular the poor and people in vvulnerable situations, including infants, to safe, nutritious and sufficient food all year round*" and "*to end all forms of malnutrition, achieving, by 2025, the internationally agreed targets on stunting and wasting in children under five years of age, and address the nutritional needs of adolescent girls, pregnant and lactating women and older persons*" [2].

In Burkina Faso, despite a 56% decline, from an estimated 202 deaths per 1,000 live births in 1990 to 89 deaths per 1,000 live births in 2015, the under-5 mortality rate did not reach the 2015 Millennium Development Goal (MDG) target of 67 per 1,000 live births [3]. In 2016, 20% of the total population was undernourished and 27% and 8% of under-5 children were estimated to suffer from stunting and wasting respectively [2].

From December 2015 to July 2017, the Alive and Thrive (A&T) initiative sought to improve breastfeeding practices in rural communes in the Boucle du Mouhoun region of Burkina Faso. The intervention included interpersonal communication activities and community mobilisation activities. The evaluation used a repeated cross-sectional cluster randomised controlled trial design, and after 14 months of full implementation, self-reported exclusive breastfeeding increased between baseline and endline surveys by 36% points more in the intervention arm compared to the control arm [4].

Baseline findings highlighted suboptimal infant and young child feeding (IYCF) practices in children 6 to 11 months old [5], but no data were collected in children aged 12 months or more. Using data collected during the endline survey in children 6 to 23 months old, the objectives of this further data analysis is to report on IYCF practices and their predictors up to the age of 2 years with a view to informing effective promotion of IYCF practices which have the potential to improve child growth [6].

## Materials and methods

We performed a cross-sectional population-based survey in June and July 2017 in the 37 rural communes (clusters) included in the cluster randomised controlled trial evaluating the A&T intervention. At the time of the survey, 18 communes had received the A&T intervention aimed at exclusive breastfeeding practices in children less than 6 months of age [5]. The study protocol and data collection tools are available to view at LSHTM Data Compass repository (https://doi.org/10.17037/DATA.280; Date last accessed 18th September 2019). Access to the dataset can also be requested through this repository.

## Setting

Boucle du Mouhoun is one of the 13 regions of Burkina Faso, located in the north-west of the country. In this region, most of the population live in rural areas, largely dependent on subsistence agriculture, and the prevalences of wasting, stunting and underweight among under-five children were estimated at 9%, 23% and 17% respectively in 2016 [7].

Boucle du Mouhoun is divided into six health districts, five with a district hospital and one with a regional hospital. In rural areas, primary health facilities, run by nurses, are the most common point of care. In all public health facilities, free antenatal care (ANC), and subsidies for childbirth and emergency obstetric and neonatal care (EmONC) are provided. At the first level of care, growth monitoring and nutrition counselling, vitamin A supplementation and deworming are provided during under-five consultations through the Integrated Management of Childhood Illness (IMCI) strategy, preventive well-baby consultations (W-BC) and outreach activities.

## Data collection

Interviews were performed in local languages (Dioula, Moore or San) using a pre-tested structured questionnaire programmed into electronic devices (using a Trimble Juno SB Personal Digital Assistant). If a mother had more than one child aged 6–23 months, twins or triplets, the youngest child was chosen as the index child. Mothers were interviewed on their knowledge, attitudes, perceived self-efficacy and practices related to IYCF, and their care seeking behaviours related to maternal and child health. For each relevant contact reported in the community or in a health facility, women were also asked whether they received information on breastfeeding or child complementary feeding.

IYCF practices were assessed using 24-hour dietary recall including a list of 29 liquids, soft, semi-solid and solid foods. Mothers were asked whether they had given their child each of the 29 items (food groups defining dietary diversity were created at the analysis stage) and were also questioned on the number of times they fed their child with soft, semi-solid or solid foods all together. Knowledge questions addressed timely introduction of seven key food groups, meal frequency and feeding practices during childhood illnesses. Attitudes towards IYCF were assessed by asking women whether they were in agreement with a set of statements. Perceived self-efficacy was assessed by asking women whether they felt capable of giving daily three key food groups to their child (meat, fish or poultry, dark green leafy vegetables, and carrot, squash or sweet potato).

The questionnaire was designed based on the questionnaires from the 2010 Burkina Faso Demographic and Health Survey (DHS) and the PROMISE trial on exclusive breastfeeding conducted in Burkina Faso [8].

The data collection involved 56 fieldworkers who were deployed in teams of six interviewers and one supervisor. Re-interviews were requested in case of incompleteness and/or inconsistencies, and all re-interviews were completed. Prior to the survey, fieldworkers received two-weeks training, including role-play in the four main languages spoken during interviews, and pilot surveys were performed in villages located outside study areas.

## Sampling procedures

Women of reproductive age, resident in the study area and mothers of a child aged 6 to 23 months living with them, were randomly selected using a two-stage sampling procedure. In each commune (cluster), three villages were first drawn with probability proportional to size using the most recent census (2006) as a sampling frame. At the second stage, eligible women living in selected villages were enumerated, and 30 mother-infant pairs were sampled per

village using simple random sampling stratified on child age group (i.e. 10 mother-infant pairs were sampled per child age group, 0–5, 6–11 and 12–23 months old). Thus, on total, per commune (cluster), were selected: 30 infants under 6 months, 30 infants aged 6 to 11 months old and 30 infants aged 12 to 23 months old.

## Sample size

The sample size was calculated for the purpose of the evaluation of the A&T intervention on exclusive breastfeeding among infants less than 6 months old [5]. A total, per cluster, of 30 mother-infant pairs where the infant is under 6 months was calculated with a view to providing at least 90% power to detect an absolute difference in exclusive breastfeeding prevalence in infants under 6 months of 50% versus 30% in intervention and control communes (clusters) respectively. We assumed a prevalence of exclusive breastfeeding of 30% prior to the trial (based on the 2010 DHS), a between-cluster coefficient of variation of 0.4 (based on a previous trial in Burkina Faso [8]) and a Type I error of 5%. The same sample size per cluster was used in older child age groups (6–11 and 12–23 months) with a view to performing descriptive analysis of IYCF prractices.

## Data analysis

All analyses were conducted using STATA/SE version 14.1. Descriptive analyses were performed using sampling weights to account for sampling stratification by child's age group and accounted for the cluster sampling approach using the svy family of commands in STATA. Although none of the indicators related to IYCF practices and reported here were targeted by the A&T initiative, balance between trial arms was checked to identify any positive effect of this intervention on IYCF practices and findings are reported overall unless imbalance between trial arms was observed.

Indicators for IYCF practices, based on the 24-hour dietary recall and defined as per WHO's guidelines [9], included: i) *Introduction of soft, semi-solid or solid foods*: proportion of children who were reported to have received at least one soft, semi-solid or solid food; ii) *Minimum dietary diversity*: proportion of children who were reported to have received foods from four or more different food groups out of seven food groups (defined as grains, roots and tubers; legumes and nuts; dairy; meat and fish; eggs; vitamin A-rich fruits and vegetables; other fruits and vegetables); iii) *Minimum meal frequency*: proportion of children who were reported to have received soft, semi-solid, or solid foods (and milk feeds for non-breastfed children) at least twice for breastfed children aged 6 to 8 months, three times for breastfed children aged 9 to 23 months, and four times for non-breastfed children aged 6 to 23 months; iv) *Minimum milk feeding frequency*: proportion of non-breastfed children who were reported to have received at least two milk feeds; v) *Minimum acceptable diet*: proportion of children who were reported to have received the minimum acceptable diet, defined as at least both the minimum dietary diversity and the minimum meal frequency for breastfed children aged 6 to 23 months, or as at least 2 milk feeds, the minimum dietary diversity (dairy excluding) and the minimum meal frequency for non-breastfed children aged 6 to 23 months.

Other indicators related to IYCF practices or key interventions related to child's growth monitoring and nutrition included: i) Child feeding practices among children whose mother reported an episode of illness in the two weeks prior to interview; ii) Zinc supplementation among children whose mother reported an episode of diarrhoea in the two weeks prior to interview; iii) Vitamin A supplementation in the six months preceding the interview; iv) Proportion of children with their height, weight and mid-upper arm circumference (MUAC) measured during their last W-BC.

All indicators were computed by age group (children aged 6 to 11 months and 12 to 23 months). As per WHO's guidelines, continued breastfeeding was also calculated at 1 year (children aged 12 to 15 months) and at 2 years (children aged 20 to 23 months), and introduction of soft, semi-solid or solid foods was also calculated in children 6 to 8 months old [9].

Univariable and multivariable logistic regression was used to identify factors predictive of IYCF indicators in children aged 6 to 23 months (introduction of soft, semi-solid or solid foods, minimum dietary diversity, minimum meal frequency and minimum acceptable diet). All regression models incorporated sampling weights to account for sampling stratification by child's age group, and commune and village as random effects to account for clustering. Variables were initially selected for inclusion based on existing literature and theory and were included in the multivariable models if associated with the respective outcome variable with p < 0.10 in the unadjusted models. For all IYCF practices, potential predictors included: at the household level, wealth quintile, clean water source and time to water source; at the mother level, age, ethnicity, religion, education level, income-generating activity (in cash or kind), partner's education level, partner's income-generating activity (in cash or kind), 4 or more ANC visit, facility delivery and postnatal care visit within one week of delivery for the pregnancy of the index child, exposure to facility-based and community-based information on complementary feeding; and at the child's level, birth order, gender, age, illness (fever, cough, fast or difficult breathing, diarrhoea) in the past two weeks, postnatal care visit within one week of brith, at least one WB-C attendance since birth, at least one visit to a health faacility for immunisation since birth. Postnatal care visit within one week of delivery either for the mother or the baby was included in the model. Household wealth index was computed from the first component of a Principal Component Analysis of 27 items collected from the household head (housing characteristics, toilet facility, agricultural land, animals and assets ownerrship). Household clean water source and time to water source were considered separately in the analysis. For introduction of soft, semi-solid or solid foods and minimum meal frequency, one additional potential predictor was tested: knowledge of timely introduction of foods and knowledge of minimum frequency respectively. A score was generated for correct knowledge of timely introduction of seven key food groups and was included as a linear term after checking for evidence of departures from linearity.

## Ethical approval

Ethical approval was granted by the National Health Ethic Committee of the Ministry of Health of Burkina Faso (Reference 2015-5-061), the institutional review board of Centre MURAZ (Reference 2015–017) and the London School of Hygiene and Tropical Medicine (Reference 9066). Written informed consent was obtained prior to interview. Findings related to ethnicities are not shown to comply with the ethical requirement in Burkina Faso. The trial is registered at ClinicalTrials.gov (Reference NCT02435524).

## Results

### Socio-demographic characteristics

Less than 1% of eligible mothers selected for interview were either absent after three visits (n = 8), unable to participate (due to sickness, cognitive or mental issues, deafness or muteness, n = 3) or refused (n = 3). A total of 2,229 women aged 15 to 49 years were interviewed, 1,116 had an infant aged 6 to 11 months and 1,113 had a child aged 12 to 23 months.

Women were 28 years old on average and predominantly of Muslim religion (63%) (S1 Table). Nearly all were in union (98%), predominantly in a monogamous union (65%).

Around 30% of women and their husbands/partners had ever attended school. On average, women had given birth to 3.9 children and their youngest children were 14 months old.

## IYCF practices

Based on mothers' reports, nearly all children (94%) were breastfed during the day or night prior to interview (Table 1). Continued breastfeeding was almost universal (99%) among children aged 12 to 15 months and remained very high (77%) among children aged 20 to 23 months. Among children reported not to be breastfed, only 9% received the minimum milk feeding frequency (Table 1).

About 70% of infants aged 6 to 11 months consumed soft, semi-solid or solid food, less than half (44%) received the minimum meal frequency, only 7% received the minimum dietary diversity, and only 6% received the minimum acceptable diet. At the age of 6 to 8 months (as per WHO guidelines for this indicator), only 54% of infants consumed soft, semi-solid or solid food. Higher proportions of children 12 to 23 months met the criteria for minimum meal frequency (68%), minimum dietary diversity (24%) and minimum acceptable diet (17%).

Fig 1 shows child feeding patterns on the day and night prior to interview by age. The proportion of children who were breastfed and consumed soft, semi-solid or solid food increased from 29% at the age of 6 months to 82% or more from the age of 9 months. From 6 to 14 months old, the proportion of children who consumed breastmilk alone or with either plain water, milk or non-milk liquids decreased from 71% to 7%.

By far, the most commonly consumed food group was grains, roots and tubers (85%) (Fig 2). Around half of children (53%) consumed vitamin A rich vegetables and fruits and about a third (37%) consumed iron rich foods (meat and fish). Each of the other food groups were consumed by about 15% or fewer children. The dietary diversity score, or mean number of food groups consumed, was 2.22 (95%CI 2.09, 2.36) in children aged 6 to 23 months: 1.33

**Table 1. IYCF practices indicators as per WHO guidelines per age group (6–11 and 12–23 months) and overall.**

| On the day and night prior to interview | 6–11 months (N = 715) | | | 12–23 months (N = 1,514) | | | 6–23 months (N = 2,229) | | |
|---|---|---|---|---|---|---|---|---|---|
| | % | 95%CI | | % | 95%CI | | % | 95%CI | |
| Introduction of soft/semi-solid/solid foods | **68.7** | 64.3 | 72.8 | **96.4** | 95.0 | 97.4 | **87.5** | 85.5 | 89.3 |
| Minimum meal frequency* | **44.4** | 39.1 | 49.7 | **67.5** | 62.3 | 72.3 | **59.8** | 54.9 | 64.5 |
| Minimum dietary diversity | **6.6** | 5.1 | 8.5 | **23.6** | 19.7 | 28.1 | **18.2** | 15.2 | 21.6 |
| Minimum acceptable diet* | **6.1** | 4.6 | 8.0 | **16.6** | 13.1 | 20.9 | **13.1** | 10.4 | 16.4 |
| | 6–11 months (N = 1) | | | 12–23 months (N = 123) | | | 6–23 months (N = 124) | | |
| | % | 95%CI | | % | 95%CI | | % | 95%CI | |
| Minimum milk feeding frequency (non-breastfed children) | **0.0** | - | - | **8.9** | 5.1 | 15.1 | **8.8** | 5.0 | 15.0 |
| During an episode of illness (2 weeks prior to interview) | 6–11 months (N = 262) | | | 12–23 months (N = 562) | | | 6–23 months (N = 824) | | |
| | % | 95%CI | | % | 95%CI | | % | 95%CI | |
| Increased breastfeeding ˚ † | **10.5** | 7.4 | 14.7 | **12.6** | 9.6 | 16.4 | **11.9** | 9.4 | 15.1 |
| Increased liquids ˚ † | **17.1** | 13.5 | 21.5 | **25.2** | 20.8 | 30.1 | **22.6** | 19.2 | 26.4 |
| Increased breastfeeding or liquids ˚ † | **22.0** | 17.8 | 26.9 | **30.3** | 25.7 | 35.2 | **27.6** | 24.1 | 31.5 |
| Continued feeding ˚ †† | **45.7** | 39.0 | 52.6 | **65.4** | 59.1 | 71.2 | **59.1** | 54.0 | 64.1 |
| Increased breastfeeding or liquids† & continued feeding††˚ | **9.3** | 6.3 | 13.4 | **19.6** | 15.5 | 24.5 | **16.3** | 13.1 | 20.2 |

* excluding 258 mothers who did not know the number of times they had given soft, semi-solid, or solid foods to their child on the day prior to interview

† Much more or more than usual

†† Much more, more, as usual or slightly less than usual

˚ 2 missing values

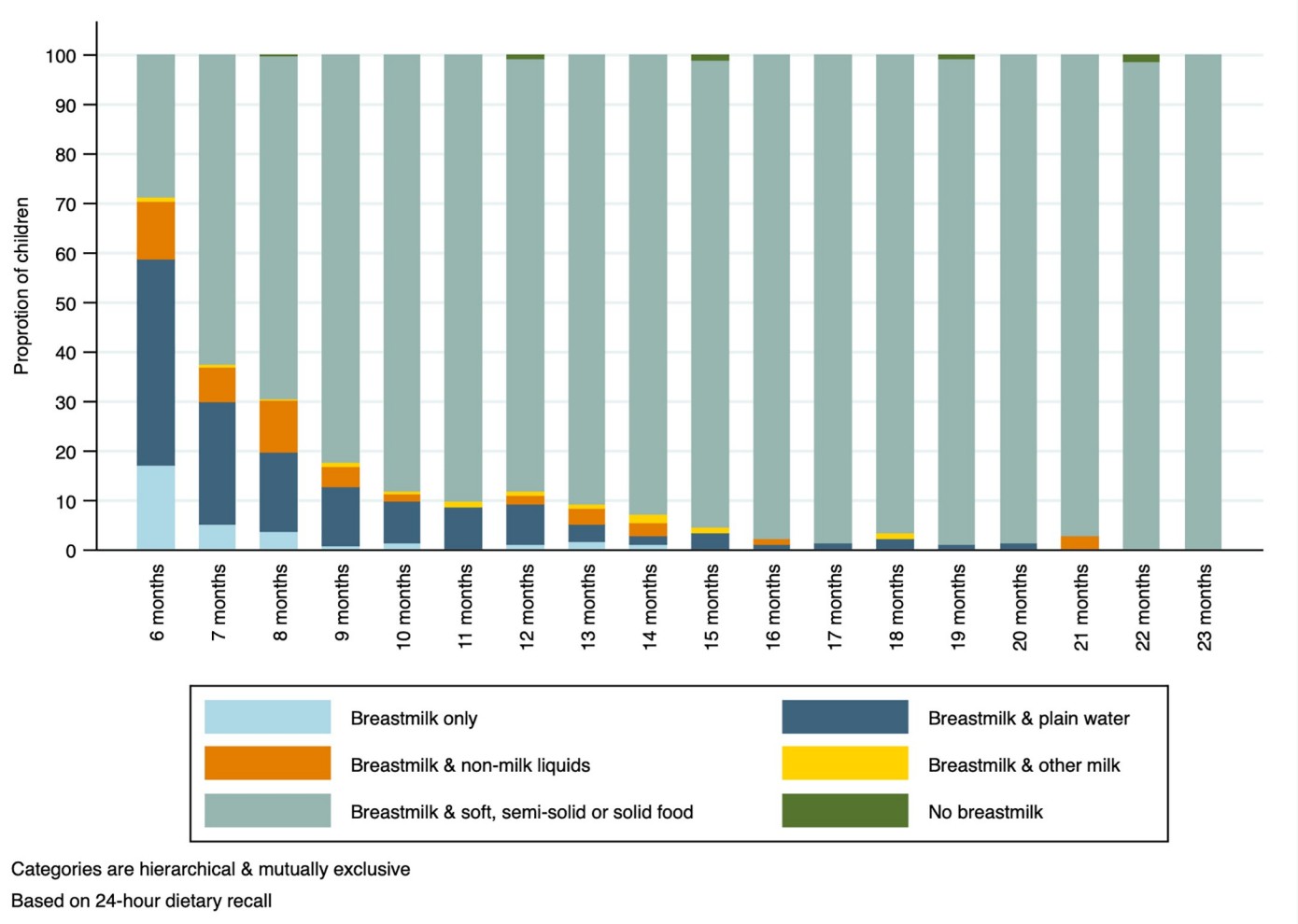

**Fig 1. Child feeding patterns by age during the day and night prior to interview (N = 2,229).**

(95%CI 1.20, 1.46) in infants aged 6 to 11 months compared to 2.65 (95%CI 2.49, 2.80) in children aged 12 to 23 months.

With respect to child feeding practices during an episode of illness, for only 9% and 20% of children aged 6 to 11 months and 12 to 23 months respectively did mothers report having increased breastfeeding or liquids and continued feeding (Table 1).

## Child growth monitoring, vitamin A and zinc supplementation

About three quarters of mothers (71%) reported having attended W-BC at least once since birth (S1 Table), but only 22% attended the last W-BC within the recommended time (within a month in children 6–11 months old and within two months in children 12–23 months old). At the last W-BC, height, weight and MUAC were reported to have been measured for 63%, 69% and 42% of children respectively (Table 2).

Despite 59% of mothers reporting having brought their child to a health facility when suffering from diarrhoea in the two weeks prior to interview (S1 Table), only a third of children (31%) received zinc supplementation (Table 2). With respect to vitamin A supplementation,

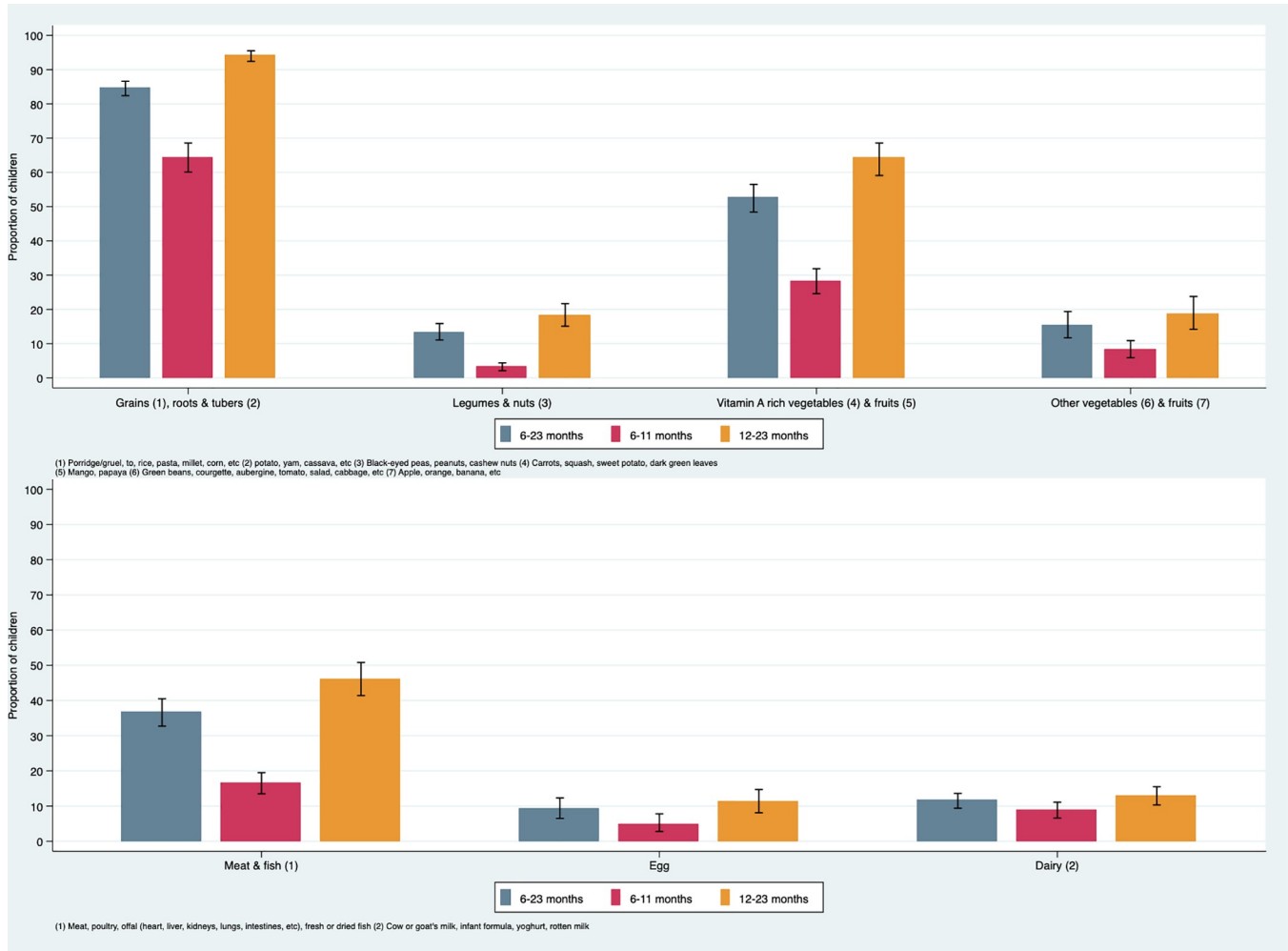

**Fig 2. Children dietary diversity and consumption of the seven food groups during the day and night prior to interview (N = 2,229).**

67% of mothers reported that their child had received a dose in the six months prior to interview.

## Knowledge, attitudes and perceived self-efficacy

With respect to knowledge, some imbalances between trial arms were observed. Higher proportions of mothers in the intervention arm correctly stated that water or other liquids (88%)

**Table 2. Child growth monitoring, vitamin A and zinc supplementation per age group (6–11 and 12–23 months) and overall.**

|  | 6–11 months (N = 715) | | | 12–23 months (N = 1,514) | | | 6–23 months (N = 2,229) | | |
|---|---|---|---|---|---|---|---|---|---|
|  | % | 95%CI | | % | 95%CI | | % | 95%CI | |
| Height measurement (last WBC) | **60.6** | 52.3 | 68.3 | **64.4** | 57.2 | 71.1 | **63.2** | 55.8 | 70.0 |
| Weight measurement (last WBC) | **66.7** | 58.4 | 74.0 | **70.0** | 63.2 | 76.0 | **68.9** | 61.9 | 75.2 |
| MUAC measurement (last WBC) | **40.8** | 32.3 | 49.8 | **42.7** | 34.9 | 50.8 | **42.1** | 34.2 | 50.4 |
| Height, weight & MUAC measurements (last WBC) | **39.3** | 30.9 | 48.5 | **41.2** | 33.6 | 49.3 | **40.6** | 32.9 | 48.9 |
| Vitamin A supplementation (past 6 months) | **65.1** | 59.5 | 70.2 | **68.6** | 63.0 | 73.6 | **67.4** | 62.2 | 72.2 |
|  | 6–11 months (N = 156) | | | 12–23 months (N = 359) | | | 6–23 months (N = 515) | | |
|  | % | 95%CI | | % | 95%CI | | % | 95%CI | |
| Zinc supplementation (children with diarrhoea, past 2 weeks) | **27.6** | 21.8 | 34.3 | **35.3** | 24.0 | 48.5 | **30.8** | 24.8 | 37.6 |

and porridge (80%) should be introduced at 6 months of age compared to 53% and 64% of mothers respectively in the control arm (Table 3). The average reported age at which water or

**Table 3. Knowledge of timely introduction of complementary foods and minimum meal frequency per trial arm and overall.**

| | | Control arm (N = 1,150) | | | Intervention arm (N = 1,079) | | | Overall (N = 2,229) | | |
|---|---|---|---|---|---|---|---|---|---|---|
| | | % | 95%CI | | % | 95%CI | | % | 95%CI | |
| **At what age should a mother start be giving her child:** | | | | | | | | | | |
| Water or other liquids? | < 6 months | **42.0** | 34.4 | 50.1 | **7.6** | 4.6 | 12.4 | **25.4** | 18.8 | 33.4 |
| | 6 months | **53.0** | 45.1 | 60.8 | **87.5** | 82.3 | 91.3 | **69.7** | 61.8 | 76.6 |
| | 7 or 8 months | **2.0** | 1.2 | 3.2 | **3.6** | 2.4 | 5.2 | **2.7** | 2.0 | 3.8 |
| | > 8 months | **0.6** | 0.2 | 1.5 | **0.7** | 0.4 | 1.5 | **0.7** | 0.4 | 1.2 |
| | Does not know | **2.4** | 1.3 | 4.5 | **0.5** | 0.3 | 1.2 | **1.5** | 0.8 | 2.7 |
| | Mean age | **4.1** | 3.6 | 4.5 | **5.8** | 5.6 | 5.9 | **4.9** | 4.5 | 5.3 |
| | Median age | **6** | - | - | **6** | - | - | **6** | - | - |
| Gruel/porridge? | < 6 months | **15.4** | 11.6 | 20.2 | **5.4** | 3.4 | 8.7 | **10.6** | 7.9 | 14.0 |
| | 6 months | **64.3** | 59.1 | 69.2 | **80.2** | 74.5 | 84.9 | **72.0** | 67.2 | 76.3 |
| | 7 or 8 months | **13.8** | 11.8 | 16.2 | **9.8** | 7.5 | 12.9 | **11.9** | 10.2 | 13.9 |
| | > 8 months | **4.7** | 2.8 | 7.8 | **3.4** | 2.2 | 5.1 | **4.1** | 2.8 | 5.8 |
| | Does not know | **1.7** | 1.2 | 2.6 | **1.2** | 0.6 | 2.1 | **1.5** | 1.0 | 2.1 |
| | Mean age | **6.1** | 5.9 | 6.2 | **6.1** | 6.0 | 6.2 | **6.1** | 6.0 | 6.2 |
| | Median age | **6** | - | - | **6** | - | - | **6** | - | - |
| Dark green leafy vegetables*? | < 6 months | **5.2** | 3.7 | 7.2 | **1.3** | 0.6 | 2.6 | **3.3** | 2.3 | 4.7 |
| | 6 months | **21.8** | 18.5 | 25.5 | **20.4** | 16.3 | 25.1 | **21.1** | 18.5 | 24.1 |
| | 7 or 8 months | **26.4** | 22.2 | 31.1 | **28.1** | 22.9 | 34.0 | **27.2** | 23.8 | 30.9 |
| | > 8 months | **43.2** | 38.2 | 48.4 | **48.2** | 43.9 | 52.5 | **45.6** | 42.2 | 49.1 |
| | Does not know | **3.4** | 2.1 | 5.3 | **2.0** | 1.2 | 3.4 | **2.7** | 1.9 | 3.9 |
| | Mean age | **8.7** | 8.4 | 9.0 | **9.1** | 8.8 | 9.3 | **8.9** | 8.7 | 9.1 |
| | Median age | **8** | - | - | **8** | - | - | **9** | - | - |
| Sweet potato? | < 6 months | **4.6** | 3.1 | 6.7 | **1.3** | 0.8 | 2.1 | **3.0** | 2.0 | 4.3 |
| | 6 months | **18.2** | 15.5 | 21.2 | **19.4** | 15.4 | 24.1 | **18.8** | 16.3 | 21.5 |
| | 7 or 8 months | **23.5** | 19.0 | 28.6 | **27.6** | 23.2 | 32.5 | **25.5** | 22.2 | 29.0 |
| | > 8 months | **48.0** | 42.4 | 53.6 | **44.9** | 41.6 | 48.3 | **46.5** | 43.2 | 49.9 |
| | Does not know | **5.8** | 4.3 | 7.8 | **6.8** | 4.7 | 9.6 | **6.3** | 5.0 | 7.9 |
| | Mean age | **9.1** | 8.9 | 9.4 | **9.0** | 8.8 | 9.2 | **9.1** | 8.9 | 9.2 |
| | Median age | **9** | - | - | **8** | - | - | **8** | - | - |
| Eggs? | < 6 months | **6.7** | 4.6 | 9.8 | **3.6** | 2.4 | 5.3 | **5.2** | 3.9 | 7.0 |
| | 6 months | **24.9** | 21.7 | 28.5 | **31.3** | 26.9 | 36.0 | **28.0** | 25.1 | 31.1 |
| | 7 or 8 months | **23.8** | 20.8 | 27.0 | **27.0** | 22.8 | 31.7 | **25.3** | 22.7 | 28.2 |
| | > 8 months | **32.9** | 28.7 | 37.4 | **26.4** | 23.2 | 29.9 | **29.8** | 26.9 | 32.8 |
| | Does not know | **11.6** | 9.0 | 14.8 | **11.7** | 9.1 | 15.1 | **11.7** | 9.8 | 13.9 |
| | Mean age | **8.3** | 8.0 | 8.6 | **8.0** | 7.8 | 8.3 | **8.2** | 8.0 | 8.4 |
| | Median age | **8** | - | - | **7** | - | - | **7** | - | - |
| Meat? | < 6 months | **3.3** | 2.2 | 4.9 | **1.4** | 0.8 | 2.2 | **2.4** | 1.7 | 3.3 |
| | 6 months | **13.3** | 10.9 | 16.0 | **13.8** | 10.8 | 17.4 | **13.5** | 11.6 | 15.7 |
| | 7 or 8 months | **17.4** | 14.3 | 21.0 | **20.4** | 18.1 | 22.9 | **18.9** | 16.8 | 21.1 |
| | > 8 months | **59.3** | 54.0 | 64.5 | **59.1** | 54.4 | 63.7 | **59.2** | 55.6 | 62.7 |
| | Does not know | **6.7** | 4.8 | 9.2 | **5.3** | 3.9 | 7.2 | **6.0** | 4.8 | 7.6 |
| | Mean age | **9.9** | 9.6 | 10.3 | **9.8** | 9.6 | 10.1 | **9.9** | 9.7 | 10.1 |
| | Median age | **10** | - | - | **10** | - | - | **10** | - | - |

*(Continued)*

**Table 3.** (Continued)

| Soft, semi-solid, solid foods? | | | | | | | | | | |
|---|---|---|---|---|---|---|---|---|---|
| | < 6 months | **5.9** | 4.1 | 8.4 | **1.4** | 0.8 | 2.6 | **3.7** | 2.6 | 5.4 |
| | 6 months | **19.6** | 16.0 | 23.7 | **17.7** | 12.7 | 24.0 | **18.6** | 15.5 | 22.3 |
| | 7 or 8 months | **24.9** | 20.8 | 29.6 | **28.4** | 23.5 | 33.7 | **26.6** | 23.3 | 30.1 |
| | > 8 months | **47.2** | 41.8 | 52.6 | **49.1** | 42.9 | 55.3 | **48.1** | 44.0 | 52.2 |
| | Does not know | **2.4** | 1.5 | 3.9 | **3.5** | 2.5 | 4.8 | **2.9** | 2.2 | 3.9 |
| | Mean age | **8.8** | 8.4 | 9.1 | **9.0** | 8.7 | 9.4 | **8.9** | 8.7 | 9.1 |
| | Median age | **8** | - | - | **9** | - | - | **8** | - | - |

| | | Control arm (N = 1,054) | | | Intervention arm (N = 1,026) | | | Overall (N = 2,080) | | |
|---|---|---|---|---|---|---|---|---|---|---|
| | | % | 95%CI | | % | 95%CI | | % | 95%CI | |
| **From that age†, how many times a day should a mother feed her child with soft/semi-solid/solid foods?** | | | | | | | | | | |
| | 2 times if 6–8 months, 3 times if 9–23 months | **63.9** | 58.0 | 69.4 | **73.8** | 66.3 | 80.1 | **68.8** | 63.8 | 73.4 |
| | Does not know | **1.8** | 1.0 | 3.3 | **1.2** | 0.7 | 2.0 | **1.5** | 1.0 | 2.3 |

* Baobab, sweet potato, cassava, black-eyed pea, moringa, spinach

† Referred to the age (6 months or above) given for the introduction of soft/semi-solid/solid foods

other liquids were introduced was 4 and 6 months in the control and intervention arms respectively, and 6 months in both arms for the introduction of porridge (with a median age at 6 months in both arms and for both water or other liquids and porridge).

Regarding other soft, semi-solid or solid foods, no substantial imbalances between trial arms were observed and 80% or more of mothers correctly stated that a child should be 6 months old or older before dark green leafy vegetables, sweet potato, eggs or meat are introduced (Table 3). Nevertheless, the mean and median age reported to introduce these soft, semi-solid or solid foods were about 9 months in both arms. In addition, among mothers who correctly stated that soft, semi-solid or solid foods should be introduced from the age of 6 months, 74% in the intervention arm had correct knowledge of minimum meal frequency compared to 64% in the control arm.

Knowledge of child feeding practices during illness was similar in both trial arms. While 85% of mothers knew that they should continue feeding their child when sick, fewer than a third correctly stated that a child should be breastfed (24%) or given liquids (30%) much more or more than usual (Table 4).

With respect to attitudes, 70% or more of mothers were in agreement with statements that a child aged 6 to 8 months old who eats egg, meat, carrots/squash/sweet potato or dark green leafy vegetables will be healthy (Table 5). However, among those who reported having introduced soft, semi-solid or solid foods, fewer felt capable of giving every day to their child dark green leafy vegetables (63%), meat, fish or egg (15%), or yellow/orange vegetables (12%).

**Table 4. Knowledge of child feeding practices during an episode of illness (N = 2,229).**

| | | % | 95%CI | |
|---|---|---|---|---|
| **When a child is sick, should a mother breastfeed her child more, as usual or less?** | "much more" or "more" | **24.4** | 21.7 | 27.3 |
| | Does not know | **0.4** | 0.2 | 0.8 |
| **When a child is sick, should a mother give her child more liquids, as usual or less?** | "much more" or "more" | **30.9** | 27.8 | 34.3 |
| | Does not know | **0.3** | 0.1 | 0.7 |
| **When a child is sick, should a mother feed her child more, as usual or less?** | "much more, "more", "as usual" or "slightly less" | **85.0** | 82.1 | 87.5 |
| | Does not know | **0.5** | 0.3 | 0.9 |

**Table 5.  Attitudes and perceived self-efficacy towards child complementary feeding.**

| Do you agree with the following statement? (N = 2,229) | | % | 95%CI | |
|---|---|---|---|---|
| "A child aged 6 to 8 months old who eats egg will be healthy" | Agree | **83.4** | 79.9 | 86.4 |
| | Does not know | **4.4** | 3.3 | 5.9 |
| "A child aged 6 to 8 months old who eats meat will be healthy" | Agree | **71.8** | 66.6 | 76.6 |
| | Does not know | **4.0** | 2.9 | 5.5 |
| "A child aged 6 to 8 months old who eats carrot/squash/sweet potato will be healthy" | Agree | **76.1** | 71.3 | 80.3 |
| | Does not know | **5.0** | 3.8 | 6.6 |
| "A child aged 6 to 8 months old who eats dark green leafy vegetables* will be healthy" | Agree | **81.1** | 77.6 | 84.2 |
| | Does not know | **2.8** | 1.9 | 3.9 |
| **Do you feel capable of giving every day to your child (mothers who introduced soft, semi-solid, solid foods, N = 1,889)?** | | % | 95%CI | |
| Meat, fish or egg | | **14.6** | 12.2 | 17.4 |
| Dark green leafy vegetables* | | **62.9** | 57.9 | 67.6 |
| carrot/squash/sweet potato | | **11.2** | 9.0 | 14.0 |

*Baobab, sweet potato, cassava, black-eyed pea, moringa, spinach

## Information on child complementary feeding

Despite relatively high proportions of mothers reporting contact with a health worker along the continuum of care (S1 Table), only 42% reported receiving information on child complementary feeding at a health facility (during a consultation for childhood illness, a visit for chid growth monitoring or for immuniation or a facility-based group discussion) in the control arm (Table 6, S1 Fig). More mothers, 66%, in the intervention arm reported receiving such information.

An imbalance between trial arms was also observed with respect to information on child complementary feeding received within the community (during a group discussion or home visit, by a local healer or a relative, by listening to radio or at other occasions) with 25% and 41% of mothers in the control and intervention arms, respectively, reporting receiving such information (Table 6, S1 Fig). Relatives most commonly reported to have given information were the woman's mother (66%), followed by her mother-in-law (48%) and a sister (17%).

**Table 6.  Information on child complementary feeding received at a health facility and in the community per trial arm and overall.**

| | Control arm (N = 1,150) | | | Intervention arm (N = 1,079) | | | Overall (N = 2,229) | | |
|---|---|---|---|---|---|---|---|---|---|
| | % | 95%CI | | % | 95%CI | | % | 95%CI | |
| At any point of care ˚ * | **41.9** | 34.3 | 49.8 | **65.6** | 57.4 | 73.0 | **53.4** | 46.5 | 60.1 |
| At any place in the community** | **24.5** | 19.2 | 30.7 | **40.6** | 34.9 | 46.6 | **32.3** | 27.5 | 37.4 |
| At any point of care* or place in the community** ˚ | **52.2** | 44.7 | 59.6 | **73.8** | 68.2 | 78.7 | **62.7** | 56.6 | 68.4 |

* During a consultation for childhood illness, a visit for child growth monitoring or for immunisation, or during a group discussion

** During a group discussion or a home visit, by a local healer or a relative, by listening to radio, or at other occasions

˚ 1 missing value

## Predictors of IYCF practices

Child's age was a strong predictor of all IYCF practices with older children more likely to be appropriately fed: For instance, 10% of children 9–11 months benefited from the minimum dietary diversity compared to only 3% of children 6–8 months (OR = 4.63, 95%CI 2.48, 8.66) (Table 7, S2–S4 Tables). Other predictors of IYCF practices included: mother's ethnicity,

**Table 7. Predictors of minimum acceptable diet (MAD) in children 6 to 23 months of age (N = 1,971).**

| | | N | MAD % | Univariable | | | | Multivariable | | | |
|---|---|---|---|---|---|---|---|---|---|---|---|
| | | | | OR | 95%CI | | P-value | OR | 95%CI | | P-value |
| Mother's age | 15–24 years | 760 | 12.7 | 1.00 | - | - | 0.536 | | | | |
| | 25–34 years | 876 | 12.8 | 1.02 | 0.75 | 1.39 | | | | | |
| | 35–49 years | 334 | 14.9 | 1.24 | 0.83 | 1.85 | | | | | |
| Mother's ethnicity* | | | | | | | 0.005 | | | | < 0.001 |
| Mother's religion | Catholic/Protestant | 574 | 10.9 | 1.00 | - | - | 0.151 | | | | |
| | Muslim | 1,254 | 14.4 | 1.26 | 0.73 | 2.15 | | | | | |
| | Animist/Atheist | 143 | 10.3 | 0.72 | 0.37 | 1.39 | | | | | |
| Mother's education level | None | 1,422 | 13.5 | 1.00 | - | - | 0.527 | | | | |
| | Primary only | 364 | 13.0 | 0.97 | 0.63 | 1.48 | | | | | |
| | Secondary or higher | 185 | 10.6 | 0.71 | 0.40 | 1.28 | | | | | |
| Mother's income generating activities (cash or kind) | No | 812 | 9.5 | 1.00 | - | - | 0.007 | 1.00 | - | - | 0.101 |
| | Yes | 1,159 | 15.6 | 1.65 | 1.15 | 2.37 | | 1.41 | 0.94 | 2.13 | |
| Mother's marital status | Monogamous union | 1,302 | 13.4 | 1.00 | - | - | 0.576 | | | | |
| | Polygamous union | 630 | 13.1 | 0.96 | 0.67 | 1.37 | | | | | |
| | Single, separated, widow | 39 | 6.9 | 0.45 | 0.10 | 2.02 | | | | | |
| Partner's education level | None | 1,299 | 13.4 | 1.00 | - | - | 0.720 | | | | |
| | Primary only | 474 | 13.0 | 1.02 | 0.66 | 1.56 | | | | | |
| | Secondary or higher | 159 | 12.7 | 1.08 | 0.55 | 2.12 | | | | | |
| | Not in union | 39 | 6.9 | 0.46 | 0.10 | 2.04 | | | | | |
| In union with a partner earning an income in cash or kind | No | 370 | 6.0 | 1.00 | - | - | 0.001 | 1.00 | - | - | 0.012 |
| | Yes | 1,601 | 14.8 | 2.69 | 1.54 | 4.69 | | 2.30 | 1.20 | 4.44 | |
| 4 or more ANC visits | No | 795 | 13.0 | 1.00 | - | - | 0.896 | | | | |
| | Yes | 1,176 | 13.2 | 0.98 | 0.67 | 1.41 | | | | | |
| Facility delivery | No | 195 | 10.4 | 1.00 | - | - | 0.928 | | | | |
| | Yes | 1,776 | 13.4 | 1.04 | 0.43 | 2.54 | | | | | |
| Postnatal care visit within 1 week of delivery (mother or baby) | No | 1,175 | 12.5 | 1.00 | - | - | 0.470 | | | | |
| | Yes | 796 | 14.1 | 1.12 | 0.82 | 1.53 | | | | | |
| Child's birth order | First live birth | 362 | 14.2 | 1.00 | - | - | 0.884 | | | | |
| | 2nd or 3rd live birth | 640 | 11.8 | 0.88 | 0.57 | 1.35 | | | | | |
| | 4th to 6th live birth | 703 | 14.0 | 1.03 | 0.75 | 1.42 | | | | | |
| | 7th or above live birth | 265 | 12.5 | 0.96 | 0.55 | 1.68 | | | | | |
| Child's gender | Boy | 1,045 | 12.5 | 1.00 | - | - | 0.354 | | | | |
| | Girl | 926 | 13.9 | 1.15 | 0.86 | 1.54 | | | | | |
| Child's age | 6–8 months | 369 | 2.8 | 1.00 | - | - | < 0.001 | 1.00 | - | - | < 0.001 |
| | 9–11 months | 284 | 10.4 | 4.81 | 2.63 | 8.78 | | 4.63 | 2.48 | 8.66 | |
| | 12–15 months | 531 | 13.3 | 6.08 | 3.69 | 10.03 | | 5.80 | 3.54 | 9.49 | |
| | 16–19 months | 446 | 19.2 | 9.70 | 5.66 | 16.62 | | 10.27 | 5.80 | 18.17 | |
| | 20–23 months | 342 | 18.3 | 9.67 | 5.20 | 17.98 | | 8.94 | 4.90 | 16.32 | |

*(Continued)*

**Table 7.** (Continued)

| | | N | MAD % | Univariable | | | Multivariable | | |
|---|---|---|---|---|---|---|---|---|---|
| | | | | OR | 95%CI | P-value | OR | 95%CI | P-value |
| Fever, cough, fast/difficult breathing or diarrhoea (past 2 weeks) | No | 1,253 | 12.8 | **1.00** | - - | 0.311 | | | |
| | Yes | 717 | 13.7 | **1.19** | 0.85 1.65 | | | | |
| At least one well-baby consultation (W-BC) attendance since birth | No | 579 | 8.7 | **1.00** | - - | 0.001 | **1.00** | - - | 0.012 |
| | Yes | 1,392 | 15.0 | **2.09** | 1.36 3.21 | | **1.93** | 1.16 3.23 | |
| At least one visit to a health facility for immunisation since birth | No | 68 | 2.9 | **1.00** | - - | 0.159 | | | |
| | Yes | 1,903 | 13.5 | **3.13** | 0.64 15.34 | | | | |
| Received facility-based information on complementary feeding | No | 910 | 10.6 | **1.00** | - - | 0.202 | | | |
| | Yes | 1,060 | 15.3 | **1.30** | 0.87 1.95 | | | | |
| Received community-based information on complementary feeding | No | 1,324 | 10.7 | **1.00** | - - | 0.001 | **1.00** | - - | 0.023 |
| | Yes | 647 | 18.1 | **1.79** | 1.26 2.53 | | **1.57** | 1.06 2.31 | |
| Household wealth quintile | Poorest | 401 | 8.5 | **1.00** | - - | 0.033 | **1.00** | - - | 0.054 |
| | Poorer | 387 | 11.4 | **1.35** | 0.72 2.52 | | **1.28** | 0.65 2.52 | |
| | Middle | 375 | 11.2 | **1.13** | 0.61 2.06 | | **0.94** | 0.51 1.75 | |
| | Richer | 393 | 16.3 | **1.92** | 1.11 3.30 | | **1.67** | 0.91 3.07 | |
| | Richest | 409 | 18.0 | **2.15** | 1.15 4.02 | | **1.97** | 1.00 3.86 | |
| Household clean water source** | No | 1,020 | 10.8 | **1.00** | - - | 0.032 | **1.00** | - - | 0.348 |
| | Yes | 951 | 15.7 | **1.46** | 1.03 2.06 | | **1.17** | 0.84 1.63 | |
| Time from water source | > 30 minutes | 389 | 11.8 | **1.00** | - - | 0.387 | | | |
| | 10 to 30 minutes | 828 | 12.1 | **1.13** | 0.72 1.77 | | | | |
| | < 10 minutes | 754 | 14.9 | **1.43** | 0.86 2.38 | | | | |

* Only P-value shown to comply with the ethical requirement in Burkina Faso

** Public fountain, borehole, tap water

mother's or partner's income generating activities, household wealth quintile, attendance to W-BC, having received information on child complementary feeding in the community and household time to water source. There was also strong evidence that having received information on child complementary feeding at a health facility and knowledge of the correct daily number of meals were associated with receiving the recommended minimum meal frequency (S4 Table).

# Discussion

According to mothers' self-reports, only 18% and 13% of children aged 6 to 23 months living in rural areas of the Boucle du Mouhoun region benefited from the minimum dietary diversity and the minimum acceptable diet respectively. These findings reflect very poor IYCF practices and are similar to those reported from the 2016 Standardized Monitoring and Assessment of Relief and Transitions (SMART) survey in the same region [7]. It should be noted, however, that the minimum dietary diversity reported at baseline in infants aged 6 to 11 months was slightly lower, at 2% [5], compared to 7% in this age group at endline. Both baseline and end-line surveys were conducted between June and July and possible explanations for this include a secular trend towards improvement, a better harvest or a more detailed dietary recall at endline.

We also found poor levels of knowledge regarding timely introduction of complementary foods. Although most mothers reported that complementary foods should be introduced from 6 months of age or older, the average reported age for introduction was 9 months.

Interestingly, mother's perceived self-efficacy to provide children with key food groups on a daily basis was also relatively low compared to generally positive attitudes towards the introduction of these food groups. Reasons for low self-efficacy were not investigated but could be related to a lack of time or money to to prepare or purchase these foods, or difficulty to find these foods on a regular basis. More information on this could have programmatic implications for effective promotion of IYCF practices.

During illness, knowledge that a child should be breastfed or given liquids more than usual was particularly poor too, and despite nearly two thirds of children with diarrhoea sought care, about a third only received zinc supplementation.

Although the A&T initiative in the Boucle du Mouhoun region primarily targeted breastfeeding practices, some messages covered complementary feeding to respond to population demands on recommendations when children reach the age of 6 months. The imbalances observed between trial arms may reflect an effect of these messages. More women in the intervention arm reported having received information on IYCF practices when visiting a health facility or within the community. While knowledge of timely introduction of eggs, meat, dark green leafy vegetables and sweet potato were balanced between arms, women in the intervention arm reported better knowledge of timely introduction of water or other liquids, gruel/porridge and minimum meal frequency compared to women in the control arm. However, reported child complementary feeding in practices did not differ between arms.

Our study has some limitations. First, our findings are based on mothers' self-reports and, as in other similar studies, recall and social desirability bias cannot be excluded. Second, the survey took place at the beginning of the rainy season when food availability can be poor. IYCF practices may be better at other times of the year, for example after the harvest. Lastly, we cannot generalise our findings to other settings in Burkina Faso given substantial differences in IYCF practices between ethnicities.

Nevertheless, our findings highlight the need for interventions to improve mothers' knowledge and practices in relation to child complementary feeding in the Boucle du Mouhoun region. We found that women who reported attending W-BC were more likely to adopt recommended IYCF practices. Although exposure to facility-based information was only predictive of child's minimum meal frequency in our study, the lack of evidence for an association with other recommended practices should be interpreted bearing in mind that neither the quality nor the frequency of information received were collected and accounted for in our analyses. The high attendance of health facilities for W-BC, albeit not as regularly as recommended, represents an opportunity for contact with caretakers that should be exploited for promotion of recommended IYCF practices and child growth monitoring. Burkina Faso, however, suffers from understaffed and under-resourced health centres and limited capacity of village health teams at the community level [10].

We also found evidence for an association of exposure to community-based information with IYCF practices, suggesting the potential of behaviour change communication strategies at the community level. Observed associations with mother's ethnicity suggests that social and cultural factors play an important role in nutrition. Qualitative data were collected during the trial and revealed that some "food taboos", such as not giving eggs to a child before s/he speaks, and a belief that solid food, if introduced too early, will delay the child's first steps may impede timely introduction of foods and dietary diversity (unpublished data). Interviews with mothers also revealed the influence of family members, in particular older women, and their role in supporting and relaying traditions has been reported in other settings, including Burkina Faso [11]. Engagement and buy-in of other family members and other community members of influence is important to ensure effective change in social norms and behaviours.

A review of randomised and non-randomised controlled trials suggests that effective promotion of IYCF practices has the potential to increase good practices and child growth with stronger evidence for an effect in food insecure populations [6]. In Bangladesh, Vietnam and Ethiopia, the A&T initiative combining facility- and/or community-based interpersonal communication, community mobilisation and mass media has been shown to be effective in improving reported IYCF practices. However, no intervention effect on child anthropometric outcomes was observed [12–15]. In Hounde district in Burkina Faso, facility-based nutrition counselling from pregnancy until 18 months after birth increased reported recommended IYCF practices and child's birth weight but did not improve other child anthropometric measures [16].Possible explanations for the lack of effect on child growth of these behaviour change communication strategies included only a small proportion of children benefiting from recommended IYCF practices at the end of the intervention, so that no effect on growth was detectable at the population level. Low health centre attendance and a relatively low coverage and intensity of the intervention was sometimes reported too. In some settings, food and resource availability may also have constrained sustained practices.

In our study, women living in poor households, who did not report an income generating activity or being in union with a partner earning an income in cash or kind were less likely to report the recommended practices. Burkina Faso ranked 185 out of 188 countries on UNDP's Human Development Index in 2014 [17], and 102 out of 118 countries on the Global Hunger Index [18] in 2016, with 16% of the population estimated to live with severe food insecurity [2]. In this context, improving feeding practices and child growth may require more practical support such as social protection interventions.

Although evidence for the effect of nutrition-sensitive programs, such as integrated agriculture-nutrition or cash transfer interventions, is inconclusive or scarce [19], this type of intervention holds promise for improving both women and children's nutrition as well as women's empowerment. In Gourma province, eastern Burkina Faso, a two-year homestead food production program with bimonthly home visits improved child feeding practices with strong evidence for an effect on diarrhoea prevalence, weak evidence for an effect on wasting and haemoglobin level, but no evidence for an effect on stunting and underweight [20, 21]. These mixed findings on child growth were discussed in light of the relatively short duration of the intervention and issues in the timing of its implementation. Some evidence for increased dietary diversity, lower prevalence of underweight and improved empowerment among mothers was also reported [22].

In Tapoa province, eastern Burkina Faso, unconditional cash transfers to mothers during the lean season increased children's intake of high-nutritional-value foods and the proportion receiving a minimum acceptable diet [23], but had no detectable effect on child growth [24]. Qualitative and household expenditure data revealed that food and health were the first two investment domains for the cash received, not only for the child, but for the whole family. The authors acknowledged the potential for a complementary behaviour change communication component to foster IYCF practices.

In conclusion, our findings revealed poor IYCF practices in rural areas of the Boucle du Mouhoun region with very few children benefiting from a minimum acceptable diet. Successful behaviour change communication strategies must be based on rigorous formative research in order to tailor the intervention to the local context. However, although these strategies have the potential to improve IYCF practices, comprehensive intervention packages that combine communication for behaviour change, increase coverage and quality of care and improvement in food security may be needed to foster and sustain change in feeding practices, and achieve optimal child growth.

## Supporting information

**S1 Table. Socio-demographic characteristics of interviewed mothers and care seeking at a health facility along the continuum of care (N = 2,229).**
(DOCX)

**S2 Table. Predictors of introduction of soft, semi-solid, solid foods (SSS) in children 6 to 23 months of age (N = 2,229).**
(DOCX)

**S3 Table. Predictors of minimum dietary diversity (MDD) in children 6 to 23 months of age (N = 2,229).**
(DOCX)

**S4 Table. Predictors of minimum meal frequency (MMF) in children 6 to 23 months of age (N = 1,971).**
(DOCX)

**S1 Fig. Information on child complementary feeding received at a health facility (left) and in the community (right) (N = 2,229).**
(DOCX)

## Acknowledgments

We gratefully acknowledge all the fieldworkers, supervisors, and data managers for their work in the field. We thank Armande Sanou and Catherine Kone who led the qualitative research and are grateful to the Alive and Thrive team for their support. We also thank the study population for their participation.

## Author Contributions

**Conceptualization:** Sophie Sarrassat, Rasmane Ganaba, Jenny A. Cresswell, Abdoulaye H. Diallo, Simon Cousens, Veronique Filippi.

**Data curation:** Henri Some, Jenny A. Cresswell.

**Formal analysis:** Sophie Sarrassat, Simon Cousens.

**Funding acquisition:** Sophie Sarrassat, Jenny A. Cresswell, Simon Cousens, Veronique Filippi.

**Methodology:** Sophie Sarrassat, Rasmane Ganaba, Henri Some, Jenny A. Cresswell, Abdoulaye H. Diallo, Simon Cousens, Veronique Filippi.

**Project administration:** Rasmane Ganaba.

**Software:** Henri Some.

**Supervision:** Sophie Sarrassat, Rasmane Ganaba, Jenny A. Cresswell, Abdoulaye H. Diallo, Simon Cousens, Veronique Filippi.

**Validation:** Rasmane Ganaba, Henri Some.

**Writing – original draft:** Sophie Sarrassat.

**Writing – review & editing:** Rasmane Ganaba, Henri Some, Jenny A. Cresswell, Abdoulaye H. Diallo, Simon Cousens, Veronique Filippi.

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
