## [Decision Letter · Decision Letter 0]

6 Sep 2019

PONE-D-19-20243

Suboptimal infant and young child feeding practices in rural Burkina Faso: Findings from a cross-sectional population-based survey

PLOS ONE

Dear Dr Sarrassat,

Thank you for submitting your manuscript to PLOS ONE. After careful consideration, we feel that it has merit but does not fully meet PLOS ONE’s publication criteria as it currently stands. Therefore, we invite you to submit a revised version of the manuscript that addresses the points raised during the review process.

We would appreciate receiving your revised manuscript by Oct 21 2019 11:59PM. To enhance the reproducibility of your results, we recommend that if applicable you deposit your laboratory protocols in protocols.io, where a protocol can be assigned its own identifier (DOI) such that it can be cited independently in the future. For instructions see: http://journals.plos.org/plosone/s/submission-guidelines#loc-laboratory-protocols

We look forward to receiving your revised manuscript.

Kind regards,

Thach Duc Tran, M.Sc., Ph.D.

Academic Editor

PLOS ONE

Journal Requirements:

1. We note that you have included the phrase “data not shown” in your manuscript. Unfortunately, this does not meet our data sharing requirements. PLOS does not permit references to inaccessible data. We require that authors provide all relevant data within the paper, Supporting Information files, or in an acceptable, public repository. Please add a citation to support this phrase or upload the data that corresponds with these findings to a stable repository (such as Figshare or Dryad) and provide and URLs, DOIs, or accession numbers that may be used to access these data. Or, if the data are not a core part of the research being presented in your study, we ask that you remove the phrase that refers to these data.

Reviewers' comments:

Reviewer's Responses to Questions

**Comments to the Author**

1. Is the manuscript technically sound, and do the data support the conclusions?

Reviewer #1: Yes

Reviewer #2: Yes

Reviewer #3: Yes

Reviewer #4: Partly

2. Has the statistical analysis been performed appropriately and rigorously? 

Reviewer #1: Yes

Reviewer #2: Yes

Reviewer #3: Yes

Reviewer #4: Yes

3. Have the authors made all data underlying the findings in their manuscript fully available?

Reviewer #1: Yes

Reviewer #2: Yes

Reviewer #3: Yes

Reviewer #4: No

4. Is the manuscript presented in an intelligible fashion and written in standard English?

Reviewer #1: Yes

Reviewer #2: Yes

Reviewer #3: Yes

Reviewer #4: Yes

5. Review Comments to the Author

Reviewer #1: General comments

This is an interesting paper and a relevant topic. Infant and child nutrition status and feeding practices are an important matter in Sub-Saharan Africa, and primarily in Sahelian countries including Burkina Faso. Nutrition issues seem more problematic in rural areas compared to cities in such settings. See below additional comments and observations aiming to help improve the manuscript.

Abstract

"2,229 mothers were interviewed" this seems more like a methodological aspect related to sample size, and not a finding related to the objective of the study. Maybe good to mention that in the methods section of the abstract when you talk about the sample.

Methods

You provided interesting arguments and justification for analyzing the nutrition theme. But, can you elaborate a bit why you decided to focus on IYCF feeding practices indicators among existing nutrition-specific indicators. Is there a scientific background or other reasons justifying such interest?

Data collection

Sampling procedures

By selecting 3 villages by rural commune, are you assuming that the communes have approximate population size? What is the rationale behind the choice of 3 villages by commune? You may have sampled the villages with a probability proportional to their size regardless the commune entity. Why you did not opt for this method? You mentioned you used the most recent census (2006) as sampling frame. Why you did not use the Enumeration areas (EAs) that are more recommended for a first stage sampling than villages?

Sample size

The sample size calculation needs to be more explicated. A few details about the procedure for the size calculation, the precision and power of the sample would be desirable. Since you are comparing the difference between the control arm and the intervention arm, it is good to ensure the sample had enough power to detect the difference between both arms for the main indicators. In the event you have information about the power-difference and level of precision of the sample, a couple of sentences in that respect would certainly reinforce the statistical reliability of the study. Or eventually, that might be stressed as potential limitation.

Second stage sampling: To double-check as the number of "20 mother-infant pairs" is indicated on row 112 and "30 mother-infant pairs" on row 134.

Data analysis

In the data analysis section, most of the variables look like explanatory variables. Did you carry out the regression models accounting for some of them as potential confounder variables? If so, good to mention that and which ones.

You mentioned that a wealth index was computed from 27 items (row 173). In table 8 for predictors of MAD, the multivariate model shows both the wealth index and household clean water source included in the model. The question is to know whether the household source water is included in the computation of the wealth index. If so, that can likely cause a multicollinearity problem since the wealth index would already be measuring that household source water is expected to measure.

Findings

Row 187-194: Suggest to add subtitle (e.g. socio-demographic characteristics or sample characteristics)

Row 193: Replace "women had had 3.9 live births" by "women had 3.9 live births"

Row 198-200: Age categories and results seems different to data in table 2. Need for double-check and correct accordingly.

Row 202-203: "(only 54% of infants aged 6 to 8 months)”. Where that comes from and why 6-8 months?

Interpretation of the knowledge results (row 236-244) not straightforward, as it is not obvious to put the analysis in perspective with data in table 4. Additionally, the mean in the table is a bit confusing. As presented, it looks like it refers to mean of the proportion of knowledge while it actually refers to the mean age. That needs clarification, and moreover a median age would be more appropriate in lieu of mean.

Table 1: "Socio-demographic characteristics of interviewed mothers and care seeking at a health facility…" can be moved as an appendix in supplementary materials, as it is a bit huge, not directly linked to the main objectives and indicators of the study, and given that there are already many tables (8) in the manuscript plus 2 figures.

Predictors of IYCF practices (row 272 and table 8). According to the bounds of 95%CI, it is a bit overstate considering the household wealth index as a predictor statistically significant.

Discussion

The discussion section is well done. It makes a good summary of the results, and well addresses the main research questions while bringing interesting literature references up for comparison, explanation or to reinforce the findings.

Good also as you included some limitations of the study. IYCF practices is indeed a season-sensitive matter and it is good that was stressed as potential limitation of the study. However, the interesting thing is to know that both baseline and endline survey were conducted during the same period (June-July) allowing for comparison.

You also tried to discuss the results according to exposure to facility-based information. I agree that such results should be discussed sparingly due to data limitation. However, it seems there is likely a missed opportunity for receiving information on child complementary feeding at a health facility, primarily in the control arm or receiving zinc supplementation for children who sought care for diarrhea. That is something you can elaborate a bit more in order to achieve the comprehension intervention package you suggested.

Formatting

Review titles of tables that are often brief (e.g. table 2, etc.) and need to be more explicit as per the content of tables

Reviewer #2: The authors report in this study on infant and young child feeding (IYCF) practices and their predictors in rural areas of Burkina Faso using results of a cross-sectional study carried out in mothers of children aged 6-23 months. The study was conducted in an area where the Alive & Thrive conducted an intervention aiming to improve breastfeeding practices. The authors found that both knowledge and actual adherence to IYCF practices were very low and showed association with children age, household and mothers’ characteristics, and previous exposure to information on child complementary feeding.

We congratulate the authors for presenting a well-structured paper, written in good English language. The overall reporting of the results was very well done, the results are sufficiently contextualized, and the authors have also comprehensively discussed the study limitations.

Below are some comments with the aim of improving the manuscript:

1. Background

Line 67 to 70: the authors are citing a secondary source (reference 2). I suggest they refer to the original statement document of the SDGs. The target 2.1 is framed as following “end hunger and ensure access by all people, in particular the poor and people in vulnerable situations, including infants, to safe, nutritious and sufficient food all year round”.

And the target 2.2 “end all forms of malnutrition, including achieving, by 2025, the internationally agreed targets on stunting and wasting in children under 5 years of age, and address the nutritional needs of adolescent girls, pregnant and lactating women and older persons”.

2. Sample size

Line 134-135: authors should carefully cross-check the number of observations per cluster, as this seems to not be consistent with what was reporting in the data collection section (line 112-113): 20 mother-infant pairs versus 30 mother-infant pairs??

Although this was not a secondary data analysis, the authors used the existing sampling frame of a previous study and the sample size arrived at was not fully justified with respect to the present study. I suggest they state the hypothesizes that underlie the sample size calculation or the compute the statistical power of the study based on the sample size and their other hypothesizes that need to be clearly stated.

3. Data analysis

Line 140-142: It is unlikely that the A&T intervention will have effect on only breastfeeding practices without any effect on other ICYF. It seems reasonable to posit some “positive externalities” that would affect the IYCF practices being assessed in the current study as the authors recognized later on in they discussion (line 295-297).

No information was provided on missingness and how it was handled and the reporting of the results do not allow the reader to assess as well missing data if any on each variable (except for the table 8 where the authors provided the absolute frequencies). My guess is that there are very few missing data, I suggest however, the authors include the “n” in the tables for more clarity and comprehensiveness.

Line 177: I suggest authors use epidemiologic variable selection in the multivariable model rather than the one based on p-values.

4. Findings

Line 187: “less than 1% of eligible mothers…”, I wonder If the authors can state the actual number and may be show a breakdown by reason of non-participation? What was the mean of “unable to participate?”. Although non participation was low, authors may still consider using a flow diagram.

Table 2: the authors should elaborate more on the title to reflect the content of the table.

Reviewer #3: Abstract: Line 43: Specify ‘60% of children received’

Introduction: The nutrition context of Burkina Faso was nicely laid out, but this section would benefit from more clearly stated aim and objectives of the paper so that readers can understand why this analysis is important for the context.

Methods:

Lines: 112 – 113: What if a mother had more than one child 6-23 months or twins/triplets?

Lines 115-116: what was the local language and were tools pretested? What was the electronic data collection system used?

Lines 117 – 125: were the questions used validated or taken from certain resources? For example, was the WHO IYCF indicator questionnaire used for adaption? And where were the self-efficacy questions developed from? More details on these tools are needed.

Line 120: how was this list of 29 liquids/foods developed? And more detail on the 24 hr recall is needed. Did mothers first list off all the foods consumed and interviewers ticked off the food items? Or was the list read out to the mothers – if so, how was comprehension of food groups assessed? (I.e. are general categories that exist in the WHO IYCF questionnaire well understood by the participants)?

Lines 133-135: Description of the sample size is not clear – what was the needed sample size, 90? And given that this was the sample size needed for the evaluation, how well/unwell powered was it to assess the prevalence of IYCF practices?

Line 145: Is this meant to be ‘Currently breastfeeding’ rather than Countinued? Continued breastfeeding at 1 year and at 2 years of age are indicators (with specific age groups for this indicator), but if these are being used please specify age group. If currently breastfed is the indicator, please note that this indicator is based on breastmilk consumption in the previous 24 hours.

Lines 160 – 164: These are not all IYCF practices, but would be better described as child nutrition indicators (particularly for the latter 2). Please also provide justification for why these additional indicators were chosen.

Line 171: What were the factors tested as predictors of IYCF indicators? Were the same predictors tested for every indicator? These are noted in lines 271 – 273 but they should be presented and each defined in the methods section. Please also provide details for why these predictors were chosen for analysis.

Lines 252 – 254: What does ‘capable’ mean? Was this related to their ability to purchase these foods? Access these foods? Have time to prepare these foods? This is an interesting finding – knowledge of healthy foods is there but a barrier is preventing the practice. More information on this barrier would be useful for programmatic implications.

Line 262: What kind of information is being described here? Health? Nutrition? And if the most common sources of this information is a relative (was there any data on receiving information from a community health worker?), do we have any idea of if this information is correct? And what specific messages were received?

Lines 292 – 293: This finding regarding self-efficacy is interesting, but a reader is left wanting more detail, as this a finding that carries implications. Was any further detail gathered regarding what the drivers of this low self-efficacy were? Any information from the A&T program operating in the communities?

Reviewer #4: The objectives of the study are not clear. According to the title, the study population should be representative of 'rural Burkina Faso'. However, only one region has been studied, of which the representativeness has not been discussed. A&T intervention areas should have been excluded, as the practices in them may deviate from those in rest of the 'rural Burkina Faso'.

A few language errors are found, especially in the Abstract section.

More comments are indicated in the attached document.

6. PLOS authors have the option to publish the peer review history of their article (what does this mean?). If published, this will include your full peer review and any attached files.

Reviewer #1: Yes: Abdoulaye Maïga, PhD

Reviewer #2: Yes: MILLOGO Tieba

Reviewer #3: No

Reviewer #4: No

---

## [Author Response · Author response to Decision Letter 0]

3 Oct 2019

Dear editor and reviewers, 

Many thanks for your comments that we hope to have taken into account properly. Our answers are in bold below and the manuscript with track changes has also been uploaded. 

Journal Requirements

Thank you for directing us to these links. We have formatted the title page, abstract, manuscript, tables, figures and supplementary information according to the guidelines provided. 

We note that you have included the phrase “data not shown” in your manuscript. Unfortunately, this does not meet our data sharing requirements. PLOS does not permit references to inaccessible data. We require that authors provide all relevant data within the paper, Supporting Information files, or in an acceptable, public repository. Please add a citation to support this phrase or upload the data that corresponds with these findings to a stable repository (such as Figshare or Dryad) and provide and URLs, DOIs, or accession numbers that may be used to access these data. Or, if the data are not a core part of the research being presented in your study, we ask that you remove the phrase that refers to these data.

“Data not shown” mentioned in the manuscript and as footnote to tables referred to findings by ethnicity groups and were not shown in order to comply with ethical requirements in Burkina Faso (see Ethical approval section in methods). We have edited the footnotes (“Only P-value shown…”) and removed the sentence “data not shown” from the manuscript. All data are accessible, and we have added at the beginning of the methods section where the protocol and the questionnaire can be viewed and access to the dataset can be requested.

 

Reviewer's Responses to Questions

1. Is the manuscript technically sound, and do the data support the conclusions?

Reviewer #1: Yes

Reviewer #2: Yes

Reviewer #3: Yes

Reviewer #4: Partly

2. Has the statistical analysis been performed appropriately and rigorously? 

Reviewer #1: Yes

Reviewer #2: Yes

Reviewer #3: Yes

Reviewer #4: Yes

3. Have the authors made all data underlying the findings in their manuscript fully available?

Reviewer #1: Yes

Reviewer #2: Yes

Reviewer #3: Yes

Reviewer #4: No

4. Is the manuscript presented in an intelligible fashion and written in standard English?

Reviewer #1: Yes

Reviewer #2: Yes

Reviewer #3: Yes

Reviewer #4: Yes

 

Review Comments to the Author

Reviewer #1

1. General comments: This is an interesting paper and a relevant topic. Infant and child nutrition status and feeding practices are an important matter in Sub-Saharan Africa, and primarily in Sahelian countries including Burkina Faso. Nutrition issues seem more problematic in rural areas compared to cities in such settings. See below additional comments and observations aiming to help improve the manuscript.

Thank you.

2. Abstract: "2,229 mothers were interviewed" this seems more like a methodological aspect related to sample size, and not a finding related to the objective of the study. Maybe good to mention that in the methods section of the abstract when you talk about the sample.

We have removed this sentence from the abstract.

3. Methods: You provided interesting arguments and justification for analyzing the nutrition theme. But, can you elaborate a bit why you decided to focus on IYCF feeding practices indicators among existing nutrition-specific indicators. Is there a scientific background or other reasons justifying such interest?

We are not sure to understand this comment. Alive and Thrive and local stakeholders were concerned about complementary feeding and wanted to be informed in more depth for future programming in this region. We used indicators recommended by WHO to provide descriptive statistics on complementary feeding in children aged 6 months or more.

4. Data collection

4.1. Sampling procedures: By selecting 3 villages by rural commune, are you assuming that the communes have approximate population size? What is the rationale behind the choice of 3 villages by commune? You may have sampled the villages with a probability proportional to their size regardless the commune entity. Why you did not opt for this method? You mentioned you used the most recent census (2006) as sampling frame. Why you did not use the Enumeration areas (EAs) that are more recommended for a first stage sampling than villages?

We have described sampling procedures in a specific sub-section in the methods. We have also recalled briefly at the beginning of the methods that the survey was conducted part of cluster randomised trial where clusters were defined by the communes of the Boucle du Mouhoun region. Thus, villages were sampled by commune/cluster. Because part of the A&T intervention (specifically the community mobilisation activities) were implemented by villages, we sampled villages, rather than EAs.

4.2. Sample size: The sample size calculation needs to be more explicated. A few details about the procedure for the size calculation, the precision and power of the sample would be desirable. Since you are comparing the difference between the control arm and the intervention arm, it is good to ensure the sample had enough power to detect the difference between both arms for the main indicators. In the event you have information about the power-difference and level of precision of the sample, a couple of sentences in that respect would certainly reinforce the statistical reliability of the study. Or eventually, that might be stressed as potential limitation.

We have provided more details on sample size calculation. Sample size was calculated for the purpose of the evaluation of the A&T intervention on exclusive breastfeeding (EBF) in infants under 6 months and with a view to providing at least 90% power to detect an absolute difference in EBF of 20% in this age group. Thus, comparisons between trail arms were only performed in relation to the objectives of the evaluation and findings are reported elsewhere (Cresswell et al, 2019). 

The same sample size was used for older children (6-11 and 12-23 months) with a view to performing descriptive analysis of IYCF practices from the age of 6 months. Because A&T intervention did not target these practices and age group, comparisons of IYCF practices between trial arms was not intended. Balance between trial arms were nevertheless checked, acknowledging that the A&T intervention may have affected other practices than EBF (see data analysis section), and unless imbalance between trial arms was observed, ICYF practices are reported overall, regardless of the trial arms.

4.3. Second stage sampling: To double-check as the number of "20 mother-infant pairs" is indicated on row 112 and "30 mother-infant pairs" on row 134.

We have described the sampling procedures and sample size calculation in more detail for better clarity and understanding of how this study was “nested” in the cluster randomised trial conducted for the purpose of the evaluation of the A&T intervention on EBF. In the previous version of the manuscript, “20 mother-infant pairs” referred to the number of pairs selected per village with 10 pairs where the infant was 6-11 months old and 10 pairs where the infant was 12-23 months old, and “30 mother-infant pairs” referred to the number of pairs selected per cluster where the infant was 0-5 months old (3 villages selected per cluster). 

5. Data analysis

5.1. In the data analysis section, most of the variables look like explanatory variables. Did you carry out the regression models accounting for some of them as potential confounder variables? If so, good to mention that and which ones.

We performed predictive modelling, not causal modelling where confounders are controlled for.

5.2. You mentioned that a wealth index was computed from 27 items (row 173). In table 8 for predictors of MAD, the multivariate model shows both the wealth index and household clean water source included in the model. The question is to know whether the household source water is included in the computation of the wealth index. If so, that can likely cause a multicollinearity problem since the wealth index would already be measuring that household source water is expected to measure.

We agree and household clean water source and time to water source were considered separately for the purpose of this analysis. We have added a sentence to clarify this.

6. Findings

6.1. Row 187-194: Suggest adding subtitle (e.g. socio-demographic characteristics or sample characteristics)

We have added a subtitle as suggested.

6.2. Row 193: Replace "women had had 3.9 live births" by "women had 3.9 live births"

We have rephrased by “Women had given birth to 3.9 children…”.

6.3. Row 198-200: Age categories and results seems different to data in table 2. Need for double-check and correct accordingly.

Continued breastfeeding is, as per WHO guidelines, defined at year 1 and year 2 and is reported in the text only, while table 2 reported “current” breastfeeding per age group (6-11 and 12-23 months). We have now removed the latter from the list of indicators reported (see data analysis section) and from table 2 (see findings section) to avoid confusion.

6.4. Row 202-203: "(only 54% of infants aged 6 to 8 months)”. Where that comes from and why 6-8 months?

The WHO indicator for the introduction of soft, semi-solid or solid foods relates to children aged 6 to 8 months, but our analyses were performed by age group 0-11 and 12-23 months and our tables are laid out accordingly (see data analysis section). Therefore, we have reported introduction of soft, semi-solid or solid foods in children 6 to 8 months in the text as well to acknowledge WHO guidelines. We have edited this sentence in the findings section for better clarity.

6.5. Interpretation of the knowledge results (row 236-244) not straightforward, as it is not obvious to put the analysis in perspective with data in table 4. 

We have tried to make it clearer.

6.6. Additionally, the mean in the table is a bit confusing. As presented, it looks like it refers to mean of the proportion of knowledge while it actually refers to the mean age. That needs clarification, and moreover a median age would be more appropriate in lieu of mean.

We have specified “mean age” in the table 4 and have added the median age as well.

6.7. Table 1: "Socio-demographic characteristics of interviewed mothers and care seeking at a health facility…" can be moved as an appendix in supplementary materials, as it is a bit huge, not directly linked to the main objectives and indicators of the study, and given that there are already many tables (8) in the manuscript plus 2 figures.

We have moved table 1 in the supplementary materials and renumbered tables in the manuscript and in the supplementary material accordingly.

6.8. Predictors of IYCF practices (row 272 and table 8). According to the bounds of 95%CI, it is a bit overstate considering the household wealth index as a predictor statistically significant.

A P-value of 0.054 represents weak evidence of an association of wealth quintile with minimum acceptable diet and we consider it is worth reporting. Also, table 2 in the appendix shows strong evidence for an association of wealth quintile with minimum dietary diversity.

7. Discussion

The discussion section is well done. It makes a good summary of the results, and well addresses the main research questions while bringing interesting literature references up for comparison, explanation or to reinforce the findings. Good also as you included some limitations of the study. IYCF practices is indeed a season-sensitive matter and it is good that was stressed as potential limitation of the study. However, the interesting thing is to know that both baseline and endline survey were conducted during the same period (June-July) allowing for comparison. You also tried to discuss the results according to exposure to facility-based information. I agree that such results should be discussed sparingly due to data limitation. However, it seems there is likely a missed opportunity for receiving information on child complementary feeding at a health facility, primarily in the control arm or receiving zinc supplementation for children who sought care for diarrhoea. That is something you can elaborate a bit more in order to achieve the comprehension intervention package you suggested.

We have highlighted the low proportion of children who received zinc supplementation during diarrhoea (despite a good frequentation of health facilities) with other key findings at the beginning of the discussion. The fact that there is missed opportunity for delivering information on child complementary feeding at health facility is included in the discussion.

8. Formatting: Review titles of tables that are often brief (e.g. table 2, etc.) and need to be more explicit as per the content of tables

We have edited the title of table 2 to better reflect its content. We also have edited titles of tables 3, 4 and 7 to indicate when results are reported either per child age group or per trial arm.

 

Reviewer #2 

The authors report in this study on infant and young child feeding (IYCF) practices and their predictors in rural areas of Burkina Faso using results of a cross-sectional study carried out in mothers of children aged 6-23 months. The study was conducted in an area where the Alive & Thrive conducted an intervention aiming to improve breastfeeding practices. The authors found that both knowledge and actual adherence to IYCF practices were very low and showed association with children age, household and mothers’ characteristics, and previous exposure to information on child complementary feeding.

We congratulate the authors for presenting a well-structured paper, written in good English language. The overall reporting of the results was very well done, the results are sufficiently contextualized, and the authors have also comprehensively discussed the study limitations.

Thank you.

Below are some comments with the aim of improving the manuscript:

1. Background

Line 67 to 70: the authors are citing a secondary source (reference 2). I suggest they refer to the original statement document of the SDGs. The target 2.1 is framed as following “end hunger and ensure access by all people, in particular the poor and people in vulnerable situations, including infants, to safe, nutritious and sufficient food all year round”. And the target 2.2 “end all forms of malnutrition, including achieving, by 2025, the internationally agreed targets on stunting and wasting in children under 5 years of age, and address the nutritional needs of adolescent girls, pregnant and lactating women and older persons”.

Reference 2 correctly states the targets 2.1. and 2.2. but we had initially shortened the statements. They are now fully quoted.

2. Sample size

2.1. Line 134-135: authors should carefully cross-check the number of observations per cluster, as this seems to not be consistent with what was reporting in the data collection section (line 112-113): 20 mother-infant pairs versus 30 mother-infant pairs?

We have added information on the sampling procedures and sample size calculation for better clarity and understanding of how this study was “nested” in the cluster randomised trial conducted for the purpose of the evaluation of the A&T intervention on exclusive breastfeeding (EBF). In the previous version of the manuscript, “20 mother-infant pairs” referred to the number of pairs selected per village with 10 pairs where the infant was 6-11 months old and 10 pairs where the infant was 12-23 months old, and “30 mother-infant pairs” referred to the number of pairs selected per cluster where the infant was 0-5 months old (3 villages selected per cluster). 

2.2. Although this was not a secondary data analysis, the authors used the existing sampling frame of a previous study and the sample size arrived at was not fully justified with respect to the present study. I suggest they state the hypothesizes that underlie the sample size calculation or the compute the statistical power of the study based on the sample size and their other hypothesizes that need to be clearly stated.

We have added information on the sample size calculation. Sample size was calculated for the purpose of the evaluation of the A&T intervention on EBF in infants under 6 months and with a view to providing at least 90% power to detect an absolute difference in EBF of 20% in this age group. The same sample size was used for older children (6-11 and 12-23 months) with a view to performing descriptive analysis of IYCF practices from the age of 6 months. Because A&T intervention did not target these practices and this age group, comparisons of IYCF practices between trial arms was not intended. 

3. Data analysis

3.1. Line 140-142: It is unlikely that the A&T intervention will have effect on only breastfeeding practices without any effect on other ICYF. It seems reasonable to posit some “positive externalities” that would affect the IYCF practices being assessed in the current study as the authors recognized later on in their discussion (line 295-297).

We have edited this sentence to acknowledge this.

3.2. No information was provided on missingness and how it was handled and the reporting of the results do not allow the reader to assess as well missing data if any on each variable (except for the table 8 where the authors provided the absolute frequencies). My guess is that there are very few missing data, I suggest however, the authors include the “n” in the tables for more clarity and comprehensiveness.

There were very few missing data. Considering the width of the tables and the very few missing data we have reported this information, when appropriate, as a footnote to the tables (see tables 1 and 6). 

Please note that the denominators of the bottom part of table 3 has changed as the correct minimum meal frequency is reported among mothers who reported 6 months or more as the age at which soft, semi-solid or solid food should be introduced.

3.3. Line 177: I suggest authors use epidemiologic variable selection in the multivariable model rather than the one based on p-values.

We agree and have clarified the methods used for inclusion of variables in the models.

4. Findings

Line 187: “less than 1% of eligible mothers…”, I wonder If the authors can state the actual number and may be show a breakdown by reason of non-participation? What was the mean of “unable to participate?”. Although non participation was low, authors may still consider using a flow diagram.

We have given details of the reason for non-participation in the text. Given the number of tables/ figures already included in the article and the very low non-participation rate, we did not consider using a flow diagram.

Table 2: the authors should elaborate more on the title to reflect the content of the table.

We have edited the title of table 2 to better reflect its content. 

 

Reviewer #3

1. Abstract: Line 43: Specify ‘60% of children received’

Thank you for spotting this. We have edited the sentence.

2. Introduction: The nutrition context of Burkina Faso was nicely laid out, but this section would benefit from more clearly stated aim and objectives of the paper so that readers can understand why this analysis is important for the context.

Thank you for highlighting this. We have now clearly stated the objectives of this analysis.

3. Methods

3.1. Lines: 112 – 113: What if a mother had more than one child 6-23 months or twins/triplets?

If a mother had more than one child aged 6-23 months, twins or triplets, the youngest child was chosen as the index child. We have clarified this in the data collection section.

3.2. Lines 115-116: what was the local language and were tools pretested? What was the electronic data collection system used?

Tools were pretested and interviews were conducted in Dioula, Moore or San. Trimble Juno SB Personal Digital Assistants were used for data collection. We have clarified this in the data collection section.

3.3. Lines 117 – 125: were the questions used validated or taken from certain resources? For example, was the WHO IYCF indicator questionnaire used for adaption? And where were the self-efficacy questions developed from? More details on these tools are needed.

The questionnaire was designed based on the questionnaires from the 2010 Burkina Faso Demographic and Health Survey (DHS) and the PROMISE trial on exclusive breastfeeding conducted in the Cascades region of Burkina Faso (Tylleskar et al, 2011). We have added this information in the data collection section. The perceived self-efficacy questions were developed with A&T.

3.4. Line 120: how was this list of 29 liquids/foods developed? And more detail on the 24 hr recall is needed. Did mothers first list off all the foods consumed, and interviewers ticked off the food items? Or was the list read out to the mothers – if so, how was comprehension of food groups assessed? (I.e. are general categories that exist in the WHO IYCF questionnaire well understood by the participants)?

The list of 29 liquids, soft, semi-solid and solid foods was developed from the questionnaires mentioned above and the pre-test. We have clarified in the data collection section that mothers were asked whether they had given their child each of the 29 items (food groups defining dietary diversity were created at the analysis stage). 

3.5. Lines 133-135: Description of the sample size is not clear – what was the needed sample size, 90? And given that this was the sample size needed for the evaluation, how well/unwell powered was it to assess the prevalence of IYCF practices?

We have added information on sample size calculation. Sample size was calculated for the purpose of the evaluation of the A&T intervention on exclusive breastfeeding in infants under 6 months and with a view to providing at least 90% power to detect an absolute difference in EBF of 20% in this age group. The same sample size was used for older children (6-11 and 12-23 months) with a view to performing descriptive analysis of IYCF practices from the age of 6 months. Because A&T intervention did not target these practices and this age group, comparisons of IYCF practices between trial arms was not intended. 

3.6. Line 145: Is this meant to be ‘Currently breastfeeding’ rather than Continued? Continued breastfeeding at 1 year and at 2 years of age are indicators (with specific age groups for this indicator), but if these are being used please specify age group. If currently breastfed is the indicator, please note that this indicator is based on breastmilk consumption in the previous 24 hours.

We have edited the data analysis section to acknowledge your comment. Reported IYCF practices were all computed based on the 24-hour dietary recall. All indicators were computed by age group (children aged 6 to 11 months and 12 to 23 months). As per WHO’s guidelines, continued breastfeeding was also calculated at 1 year (children aged 12 to 15 months) and at 2 years (children aged 20 to 23 months), and introduction of soft, semi-solid or solid foods was also calculated in children 6 to 8 months old.

Because continued breastfeeding is specifically reported at 1 year and at 2 years it is not included in the table 2 but reported in the findings section (subsection ICYF practices). We removed “Current” breastfeeding (incorrectly labelled “continued breastfeeding”) from table 2 as it is, to some extent, redundant with continued breastfeeding;

3.7. Lines 160 – 164: These are not all IYCF practices but would be better described as child nutrition indicators (particularly for the latter 2). Please also provide justification for why these additional indicators were chosen.

We agree and have edited the sentence. These indicators are also about child’s nutrition and their importance is highlighted in the IMCI guidelines (feeding practices, in particular increase liquids and zinc supplementation in children with diarrhoea and vitamin A supplementation).

3.8. Line 171: What were the factors tested as predictors of IYCF indicators? Were the same predictors tested for every indicator? These are noted in lines 271 – 273 but they should be presented, and each defined in the methods section. Please also provide details for why these predictors were chosen for analysis.

We have developed this paragraph of the data analysis section. The same potential predictors were tested for introduction of soft, semi-solid or solid foods, minimum dietary diversity, minimum meal frequency and minimum acceptable diet. Variables were initially selected for inclusion based on existing literature and theory. For introduction of soft, semi-solid or solid foods and minimum meal frequency, one additional potential predictor was tested: knowledge of timely introduction of foods and knowledge of minimum frequency respectively. 

4. Results

4.1. Lines 252 – 254: What does ‘capable’ mean? Was this related to their ability to purchase these foods? Access these foods? Have time to prepare these foods? This is an interesting finding – knowledge of healthy foods is there but a barrier is preventing the practice. More information on this barrier would be useful for programmatic implications.

The question was formulated as such, i.e. “Do you feel capable of giving every day to your child: Meat, fish or egg?; Baobab, sweet potato, cassava, black-eyed pea, moringa, spinach?; carrot/squash/sweet potato?” (table 6). Reasons for low self-efficacy were not investigated and we have acknowledged your comment in the discussion section.

4.2. Line 262: What kind of information is being described here? Health? Nutrition? And if the most common sources of this information is a relative (was there any data on receiving information from a community health worker?), do we have any idea of if this information is correct? And what specific messages were received?

Information on child complementary feeding is described in this paragraph. This is now clarified as well as where this information was received within the community. Figure 1 in the supplementary material gives the proportion of mothers who reported to have received information on child complementary feeding during a group discussion, during a home visit, by a local healer, by a relative, by listening to radio and at other occasions. The most common source of information within the community was the radio, and in the intervention arm, group discussion (likely reflecting the community mobilisation component of the A&T intervention). The fact that neither the quality nor the frequency of information received were collected is acknowledged in the discussion section.

5. Discussion

5.1. Lines 292 – 293: This finding regarding self-efficacy is interesting, but a reader is left wanting more detail, as this is a finding that carries implications. Was any further detail gathered regarding what the drivers of this low self-efficacy were? Any information from the A&T program operating in the communities?

The reasons why mothers expressed low self-efficacy in providing key food groups to their child on a daily basis were not investigated. We have acknowledged your comment in the discussion section and have highlighted the need to gain more insights on this to inform effective promotion of IYCF practices.

Reviewer #4 

1. The objectives of the study are not clear. 

We have clarified the objectives of the study.

2. According to the title, the study population should be representative of 'rural Burkina Faso'. However, only one region has been studied, of which the representativeness has not been discussed. 

We have edited the title and specified “in rural Boucle du Mouhoun, Burkina Faso”.

3. A&T intervention areas should have been excluded, as the practices in them may deviate from those in rest of the 'rural Burkina Faso'.

We agree that the A&T intervention, although targeting breastfeeding practices in infants aged 0-5 months old, may have impacted IYCF practices in older children. This is acknowledged in the manuscript and imbalance between trial arms were checked and reported when occurring (knowledge of timely introduction of water and other liquids, and porridge, as well as information received on complementary feeding). 

4. A few language errors are found, especially in the Abstract section. 

Thank you. We have edited the abstract according to your comments.

5. More comments are indicated in the attached document.

Please see the attached document where we have added our answers to your comments.

---

## [Decision Letter · Decision Letter 1]

22 Oct 2019

Suboptimal infant and young child feeding practices in rural Boucle du Mouhoun, Burkina Faso: Findings from a cross-sectional population-based survey

PONE-D-19-20243R1

Dear Dr. Sarrassat,

We are pleased to inform you that your manuscript has been judged scientifically suitable for publication and will be formally accepted for publication once it complies with all outstanding technical requirements.

With kind regards,

Thach Duc Tran, M.Sc., Ph.D.

Academic Editor

PLOS ONE

Additional Editor Comments (optional):

Reviewers' comments:

Reviewer's Responses to Questions

**Comments to the Author**

1. If the authors have adequately addressed your comments raised in a previous round of review and you feel that this manuscript is now acceptable for publication, you may indicate that here to bypass the “Comments to the Author” section, enter your conflict of interest statement in the “Confidential to Editor” section, and submit your "Accept" recommendation.

Reviewer #1: All comments have been addressed

Reviewer #2: All comments have been addressed

Reviewer #3: All comments have been addressed

2. Is the manuscript technically sound, and do the data support the conclusions?

Reviewer #1: Yes

Reviewer #2: Yes

Reviewer #3: Yes

3. Has the statistical analysis been performed appropriately and rigorously? 

Reviewer #1: Yes

Reviewer #2: Yes

Reviewer #3: Yes

4. Have the authors made all data underlying the findings in their manuscript fully available?

Reviewer #1: (No Response)

Reviewer #2: Yes

Reviewer #3: Yes

5. Is the manuscript presented in an intelligible fashion and written in standard English?

Reviewer #1: (No Response)

Reviewer #2: Yes

Reviewer #3: Yes

6. Review Comments to the Author

Reviewer #1: My main comments were related to sample size calculation and sampling procedures. Both issues have been well addressed and revisions made accordingly.

I understand that you have used villages as cluster because "A&T intervention were implemented by villages". Although this is a tenable argument, you may have used Enumeration areas (EAs) instead. EAs were available for the study setting, total population size is quite similar per unit using EA, and EAs are more appropriate as statistical geographical unit with clear boundaries, and for better representativeness of the population. Although the village can be used as sampling unit (with related limitations), best to consider using EA (if relevant, available, affordable and accurate) for future studies.

Reviewer #2: I would like to congratulate authors one more time for their paper and for providing more clarity in the current version which I found satisfactory

Reviewer #3: (No Response)

7. PLOS authors have the option to publish the peer review history of their article (what does this mean?). If published, this will include your full peer review and any attached files.

Reviewer #1: Yes: Abdoulaye Maïga

Reviewer #2: Yes: Millogo Tieba

Reviewer #3: Yes: Alissa Pries

---

## [Editor Report · Acceptance letter]

5 Nov 2019

PONE-D-19-20243R1 

Suboptimal infant and young child feeding practices in rural Boucle du Mouhoun, Burkina Faso: Findings from a cross-sectional population-based survey 

Dear Dr. Sarrassat:

I am pleased to inform you that your manuscript has been deemed suitable for publication in PLOS ONE. Congratulations! Your manuscript is now with our production department. 

With kind regards,

on behalf of

Dr. Thach Duc Tran 

Academic Editor

PLOS ONE